# *Mycobacterium tuberculosis* infection, immune activation, and risk of HIV acquisition

**Rachel A. Bender Ignacio**[1,2]*, **Jessica Long**[1], **Aparajita Saha**[1], **Felicia K. Nguyen**[1], **Lara Joudeh**[1], **Ethan Valinetz**[1], **Simon C. Mendelsohn**[3], **Thomas J. Scriba**[3], **Mark Hatherill**[3], **Holly Janes**[2], **Gavin Churchyard**[4,5,6], **Susan Buchbinder**[7], **Ann Duerr**[2], **Javeed A. Shah**[1,8], **Thomas R. Hawn**[1]

**1** Department of Medicine, University of Washington, Seattle, WA, United States of America, **2** Vaccine and Infectious Diseases Division, Fred Hutchinson Cancer Research Center, Seattle, WA, United States of America, **3** South African Tuberculosis Vaccine Initiative, Institute of Infectious Disease and Molecular Medicine, and Division of Immunology, Department of Pathology, University of Cape Town, Cape Town, South Africa, **4** Aurum Institute, Parktown, South Africa, **5** School of Public Health, University of Witwatersrand, Johannesburg, South Africa, **6** Department of Medicine, Vanderbilt University, Nashville, TN, United States of America, **7** San Francisco Department of Public Health and Departments of Medicine and Epidemiology, University of California San Francisco, San Francisco, CA, United States of America, **8** Veteran Affairs Puget Sound Healthcare System, Seattle, WA, United States of America

* rbi13@uw.edu

**Data Availability Statement:** The full dataset for this study is available from the Github repository (https://github.com/jesslong1/HVTN-502-504-LTBI-Dataset).

## Abstract

### Background

Although immune activation is associated with HIV acquisition, the nature of inflammatory profiles that increase HIV risk, which may include responses to *M. tuberculosis* (*Mtb*) infection, are not well characterized.

### Methods

We conducted a nested case-control study using cryopreserved samples from persons who did and did not acquire HIV during the multinational Step clinical trial of the MRKAd5 HIV-1 vaccine. PBMCs from the last HIV-negative sample from incident HIV cases and controls were stimulated with *Mtb*-specific antigens (ESAT-6/CFP-10) and analyzed by flow cytometry with intracellular cytokine staining and scored with COMPASS. We measured inflammatory profiles with five Correlates of TB Risk (CoR) transcriptomic signatures. Our primary analysis examined the association of latent *Mtb* infection (LTBI; IFNγ+CD4+ T cell frequency) or RISK6 CoR signature with HIV acquisition. Conditional logistic regression analyses, adjusted for known predictors of HIV acquisition, were employed to assess whether TB-associated immune markers were associated with HIV acquisition.

### Results

Among 465 participants, LTBI prevalence (21.5% controls vs 19.1% cases, p = 0.51) and the RISK6 signature were not higher in those who acquired HIV. In exploratory analyses, *Mtb* antigen-specific polyfunctional CD4+ T cell COMPASS scores (aOR 0.96, 95% CI 0.77, 1.20) were not higher in those who acquired HIV. Two CoR signatures, Sweeney3 (aOR

**Funding:** This research was funded in large part by a 2018 CFAR grant from the University of Washington / Fred Hutch Center for AIDS Research, an NIH funded program under award number AI027757 which is supported by the following NIH Institutes and Centers: NIAID, NCI, NIMH, NIDA, NICHD, NHLBI, NIA, NIGMS, NIDDK. The parent study was supported by Merck Research Laboratories and the US National Institutes of Health (grant UM1 AI068614) and the HVTN SDMC grant UM1AI06863. Additional support from grants from NIH/NIAID K23AI129659 to RBI and 2K24AI137310 to TRH; SCM is a recipient of PhD funding from the Fogarty International Center of the NIH under Award Number D43 TW010559, the Harry Crossley Clinical Research Fellowship, and the South African Medical Research Council (SAMRC) through its Division of Research Capacity Development under the SAMRC Clinician Researcher Programme. The funders had no role in study design, data collection and analysis, decision to publish, or preparation of the manuscript.

**Competing interests:** I have read the journal's policy and the authors of this manuscript have no competing interests. RBI is a scientific consultant to SeaGen and AbbVie for unrelated work and declares that due to the unrelated area of research, there are no competing interest, but the consulting work is presented for full disclosure.

1.38 (1.07, 1.78) per SD change) and RESPONSE5 (0.78 (0.61, 0.98)), were associated with HIV acquisition. The transcriptomic pattern used to differentiate active vs latent TB (Sweeney3) was most strongly associated with acquiring HIV.

## Conclusions

LTBI, *Mtb* polyfunctional antigen-specific CD4+ T cell activation, and RISK6 were not identified as risks for HIV acquisition. In exploratory transcriptomic analyses, two CoR signatures were associated with HIV risk after adjustment for known behavioral and clinical risk factors. We identified host gene expression signatures associated with HIV acquisition, but the observed effects are likely not mediated through *Mtb* infection.

## Background

Several lines of evidence suggest that systemic inflammation increases risk of HIV acquisition. For example, in the HIV Vaccine Trials Network Step Study (V520-023/HVTN 502), an efficacy trial of an Adenovirus 5 (Ad5)-vectored HIV-1 vaccine, some vaccinees experienced a *transient* increased risk of HIV acquisition for several months after vaccination [1, 2]. In Step, ELISpot mock responses (interferon-gamma [(IFN-] secretion in the absence of antigen), but not HIV-antigen stimulated responses, directly correlated with a 61% increase in HIV acquisition risk [3]. Furthermore, CAPRISA-004 and Partners in Prevention studies each identified different cytokine profiles associated with HIV acquisition [4, 5]. Other studies suggest associations between herpes simplex virus-2 and chronic filarial infections with HIV acquisition [6–8]. Although there is some understanding of the local genital inflammation and microbiome changes associated with HIV susceptibility [9–11], the biologic underpinnings of peripheral blood immune profiles potentially associated with HIV risk are not well understood. It may be that other infections or exposures induce immune phenotypes that increase susceptibility of CD4 T cells and macrophages to HIV infection.

*Mycobacterium tuberculosis* (*Mtb*) latently infects approximately 24% of the world's population, with an estimated 10 million new cases of clinical disease (TB) in 2019 alone [12–14], Latent *Mtb* infection (LTBI), is defined by a positive response to *Mtb* antigens (Purified Protein Derivative (PPD) skin test or an IFN- release assay (IGRA)) in the absence of clinical symptoms. Recent data indicates that many asymptomatic individuals have "incipient TB" with evidence of systemic inflammation [15], which is associated with an increased risk of progression to clinical TB disease [16, 17]. While the effects of HIV on risk of LTBI progression have been intensively investigated, the effect of *Mtb* infection on risk of acquiring HIV has never been formally assessed, other than in ex-vivo experiments [18–21]. In a recent study, infant macaques were vaccinated with TB vaccines and then challenged with simian immunodeficiency virus (SIV) orally. Animals vaccinated with either an *Mtb*-SIV auxotroph vaccine or BCG vaccine demonstrated significantly increased SIV risk compared to placebo recipients. Both vaccines resulted in immune activation, which correlated with SIV acquisition [22].

Because the global burden of *Mtb* infection is high and the lifetime risk of progression to active disease is only ~10% in HIV-negative persons, no countries with co-endemic TB and HIV routinely treat LTBI in HIV-negative persons [23]. If treating LTBI were to reduce the risk of acquiring HIV, then targeting LTBI treatment to the subset of individuals most at-risk for HIV could prevent development of TB disease as well as decrease risk of HIV. We therefore designed a study of LTBI- and *Mtb*-associated immune activation using prospectively collected

samples from an HIV vaccine trial to address this question. We included several Correlates of Risk (CoR) transcriptomic signatures that have been validated to predict TB-disease states, including incipient active TB and TB treatment success, based on differential human gene activation.

## Methods

### Study population

We conducted a case control ancillary study nested within the completed Step study (V520-023/HVTN502). The Step study enrolled 3,000 high-risk HIV-negative males and females aged 18–45 who received the MRKAd5 HIV-1 gag/pol/nef vaccine at 34 study sites in the Americas, Caribbean, and Australia from 2004–2007. Participants received vaccine/placebo injection at 0, 4, and 26 weeks. HIV testing was done at 6-monthly visits for up to 4 years of in the Step study and the follow-on HVTN 504; stored blood from the prior visit was tested for HIV RNA to more precisely time HIV acquisition [2]. A total of 172 incident infections were diagnosed in men (3.3/100 person-years) and 15 infections in women (0.45/100 PY) [1]. We identified as cases all participants within Step and HVTN 504 with incident HIV who had available peripheral blood mononuclear cells (PBMC) at the last study visit prior to HIV acquisition. Cases who acquired HIV before week 8 (first available PBMC) were excluded. Controls were selected by HVTN staff as persons who did not acquire HIV during study follow-up, matched to cases 2:1 on study site and treatment arm, and with a pooled gender distribution similar to cases. We acquired PBMC from the study visit corresponding to that of their matched case; if no sample from a matched visit was available a new control was selected. An unblinded team member not involved in the laboratory work constructed a batching scheme that allowed for a balanced selection of cases and controls throughout the laboratory work, while allowing our laboratory team to remain blinded to participant characteristics. The following baseline variables from the Step study were assessed: age, gender, self-reported race/ethnic identity, and study site; study treatment (vaccine/placebo); baseline Ad5 antibody titer, self-reported behavioral HIV risks: drug use, number and type of sexual partners, condomless sex, substance use; circumcision status and HSV-2 serostatus for males. Step participants were generally not eligible for TB prophylaxis and no prophylactic or therapeutic TB drugs were documented as given to any participant during study follow up. Because this ancillary study used only deidentified data and samples from the Step trial, the work described in this manuscript was determined not to be engaged in human subjects research by the University of Washington Institutional Review Board. All participants who participated in the Step study and whose samples and data were used here were verified to have given specific informed consent for future use of stored samples at time of participation in the vaccine trial.

### Cell activation and flow cytometry

Cryopreserved PBMCs were thawed, washed and rested in RPMI 1640 media containing 10% heat-inactivated fetal bovine serum (FBS) overnight prior to antigen stimulation with a concentration of $2 \times 10^6$ cells /mL. PBMCs were counted with Guava easyCyte (Millipore) using Guava ViaCount reagent (Luminex) and GuavaSoft v.2.6 software. Samples with less than 66% viability were discarded. Cells were stimulated with a pool of early secretory *Mtb* antigen target-6 (ESAT-6) and culture filtrate protein (CFP-10) (BEI Resources). PMA (25ng/mL)/ionomycin (1ug/mL) was used as a positive control and DMSO (0.5%) was used as a negative control. In addition, costimulatory antibody anti-CD28/49d, cytokine secretion inhibitor Brefeldin A and Monensin were added to each stimulation cocktail. Cells were lysed and permeabilized with FACS Lyse and FACS Perm-II buffer and stained on a BD LSRFortessa with an

antibody panel developed for analyzing CD4 and CD8 T-cell responses (ICS) including IFN-, TNF, IL-2, and IL-17a (**Table A in S1 Appendix**) [24]. Flow cytometry data were analyzed in FlowJo™ v10.7. Each sample was compensated and gated manually.

## RNA isolation and RT-PCR

Approximately 2 x 10$^6$ PBMCs from each sample were re-suspended in RNA*later* (Invitrogen) preservative and stored at -20˚C for processing [25]. Total RNA was isolated using RNeasy spin columns (Qiagen) and cDNA templates for qRT-PCR were generated using Applied Bio-systems high-capacity cDNA reverse transcriptase kits. qRT-PCR was performed with 44 primer-probe sets (**Table B in S1 Appendix**) from Taqman using the Fluidigm 96.96 dynamic array platform. cDNA was preamplified using a pool of specific TaqMan primer-probe sets for 16 cycles with a 15 second denaturation step at 95˚C and 4 minutes at 60˚C. The pre-amplified reaction was added to the Fluidigm 96.96 dynamic array platform with TaqMan gene expression assays and amplified for an additional 40 cycles. A positive control sample was run on every chip to monitor consistency. Primer-probe sets were included to encompass the components of 5 previously described CoR signatures from peripheral blood associated with TB disease outcomes. Signature scores were calculated from raw Ct values in R v3.6.1 as previously described [25–30] (**Table B in S1 Appendix**; additional primer probes not pertaining to selected CoRs were included for future analysis).

## Statistical analysis

The outcome of interest in all analyses was HIV acquisition. To evaluate the association of *Mtb* antigen-specific T-cell activation and HIV acquisition, we used three measures:

1. Individuals with LTBI were defined using a flow cytometry-based method that is strongly correlated with the tuberculin skin test and IGRA. Samples were classified as positive if the frequency of IFN-+ CD4+ cells stimulated with ESAT-6/CFP-10 pooled peptides doubled compared to negative control [31–35]. LTBI was prespecified as the primary *Mtb* exposure variable.

2. We used **Com**binatorial **p**olyfunctionality analysis of **a**ntigen-**s**pecific T-cell **s**ubsets (COMPASS) to determine overall *Mtb*-antigen-specific T cell activation. COMPASS is an analytic tool that uses a Bayesian hierarchical framework to model all observed cell subsets. COMPASS outputs functional scores (FS) and polyfunctional scores (PFS) that define the posterior probabilities of antigen-specific response across cell subsets, described by a single numerical score that ranges from 0 to 1 [36]. We have previously used COMPASS to investigate polyfunctionality in CD4+ T cells in an epidemiologic study of South African adolescents screened for LTBI, as well as to predict HIV-specific responses in a prior HIV vaccine trial [37, 38]. FS and PFS derived from COMPASS output were included as untransformed variables of secondary interest.

3. To determine whether transcriptomic evidence of *Mtb*-associated immune activation also predicts risk of HIV acquisition, we used Correlates of Risk (CoR) scores previously validated to detect different TB-associated states. The primary signature of interest was RISK6, a 6-gene transcriptomic CoR signature that predicts progression to TB disease (incipient TB) [26]. Additional CoR scores assessed in exploratory analysis include Suliman4 and Maertzdorf4, both of which also predict incipient TB; Sweeny3, which accurately differentiates between current LTBI vs active TB; and RESPONSE5, which predicts cure at time of TB treatment initiation among those with active TB (**Table B in S1 Appendix**) [27–30]. CoR scores that were not normally distributed were log$_{10}$ transformed (RISK6, and

Suliman4). To account for score values of zero (n = 3; all RISK6), zeros were replaced by dividing the next lowest value by two.

For all measures, the association between exposures and HIV acquisition was estimated using univariate and multivariate conditional logistic regression accounting for the variables used to match controls to cases (country of residence and treatment arm within the Step study). Variables assessed for inclusion in multivariate regression included known predictors of HIV risk in the Step study (i.e. gender, validated behavioral risk score (scale 0–7) [39], baseline Ad5 titer, and self-reported race/ethnicity). Each variable of interest was assessed for association with HIV acquisition using conditional logistic regression, and associations that were significant at p<0.1 were included in the multivariate model. For sub-analyses restricted to males, we included HSV-2 serostatus and circumcision, as these were documented in the parent trial for males only. Analyses of the association of PFS, FS, and CoR scores with HIV acquisition were repeated after stratification by LTBI status. Stratified analyses were performed as standard logistic regression with matching variables included, as logistic regression conditioned on LTBI would have excluded all case/control sets with discordant LTBI status. PFS, FS, and CoR scores were reported as the Odds Ratio (OR) for 1 standard deviation change to allow for comparability across outcomes. As a sensitivity analysis, we repeated the primary analyses restricted to case/control sets with samples within 6 and 12 months prior to HIV diagnosis. All analyses were conducted in R v3.6.1 and Stata v15.1 (College Station, TX, USA, 2017).

## Results

We selected 155 cases who acquired HIV and 310 controls (n = 465), with baseline characteristics presented in **Table 1**. Among cases, the median interval between sample collection date and HIV diagnosis was 287 days (IQR 161, 483). The majority of participants (88%) were from Peru or the US. By design, there was no difference in distribution of vaccine treatment arm or country in cases vs. controls, although there was a trend toward more males among cases. As expected, the behavioral risk score previously found to be associated with HIV acquisition was higher in cases than controls (3.3 vs 2.8, SD 1.2, p<0.0001), and HSV-2 seropositivity was more common in male cases than controls (40.1% vs 30.7%, p = 0.049).

To assess one of our primary hypotheses, whether *Mtb* infection was associated with HIV acquisition, we examined 454 participants, including 152 cases and 302 controls (n = 11 missing samples or gating failures) for LTBI status via flow cytometry. Overall, 94 (20.7%) study participants were considered LTBI positive via detection of ESAT-6-CFP-10-specific CD4+ T cells: 65 (21.5%) controls compared to 29 (19.1%) cases. LTBI positivity was not associated with HIV acquisition in unadjusted or adjusted conditional logistic regression analyses (aOR 0.85 (0.51, 1.43) per 1 SD change, p = 0.73) (**Table 2**). When stratified by gender, the inference did not change. Among men, when adjusting for all covariates in the primary models as well as HSV-2, LTBI status was not associated with HIV acquisition (aOR: 0.99; 95% CI 0.58, 1.69). Likewise, there was no association between LTBI status and HSV-2 serostatus (31.8% seropositive in LTBI negative vs 30.0% in LTBI positive, p = 0.77).

We also performed two exploratory analyses with ESAT-6/CFP-10-specific functional and polyfunctional cytokine expression, as measured through the COMPASS FS and PFS, respectively, which were both significantly higher in LTBI-positive versus LTBI-negative samples (p<0.0001; **Fig 1A**). Among all 152 cases the median FS was 0.026 (IQR 0.009, 0.077) and of 302 controls the median FS was 0.029 (0.009, 0.077). In controls, the median PFS was 0.015 vs 0.015 among cases and controls. **Fig 1B and 1C** shows box plots and a heatmap of COMPASS posterior probabilities by case/control status. The distributions of mono- and poly-functional

**Table 1. Participant characteristics.**

| Characteristic | Control (n = 310) | Case (n = 155) |
|---|---|---|
| **Age (mean, SD)** | 31.9 (7.8) | 30.6 (7.8) |
| **Gender** | | |
| **Male** | 264 (85.2%) | 142 (91.6%) |
| **Female** | 46 (14.8%) | 13 (8.4%) |
| **Country** | | |
| **Australia** | 4 (1.3%) | 2 (1.3%) |
| **Brazil** | 12 (3.9%) | 6 (3.9%) |
| **Canada** | 14 (4.5%) | 7 (4.5%) |
| **Dominican Republic** | 2 (0.6%) | 1 (0.6%) |
| **Haiti** | 4 (1.3%) | 2 (1.3%) |
| **Peru** | 68 (21.9%) | 34 (21.9%) |
| **USA** | 206 (66.5%) | 103 (66.5%) |
| **Race and Ethnicity***  | | |
| **White** | 152 (49.0%) | 79 (51.0%) |
| **Black** | 43 (13.9%) | 15 (9.7%) |
| **Multiracial or Other Race** | 12 (3.9%) | 11 (7.1%) |
| **Mestizo/a** | 68 (21.9%) | 34 (21.9%) |
| **Hispanic** | 35 (11.3%) | 16 (10.3%) |
| **Vaccine trial treatment arm** | | |
| **Intervention** | 177 (57.1%) | 86 (55.5%) |
| **Comparison** | 133 (42.9%) | 69 (44.5%) |
| **AD5 Titer stratum** | | |
| **>18** | 176 (56.8%) | 84 (54.2%) |
| **< = 18** | 134 (43.2%) | 71 (45.8%) |
| **Behavioral Risk score (mean, SD)**** | 2.8 (1.2) | 3.3 (1.2) |
| **Circumcised (male participants)** | 157 (59.5%) | 84 (59.2%) |
| **HSV-2 seropositive (male participants)** | 81 (30.7%) | 57 (40.1%) |

*Race and ethnicity were asked in a singular question about self-identity in the parent study without option of multiselect; options including race or ethnicity therefore sum to 100%.

**Combined risk score for male and female includes theoretical and observed range 0–7 points.

*Mtb*-specific subsets across persons who did or did not acquire HIV were similar. When assessing the association using conditional logistic regression, neither FS (OR 0.94 per 1 SD change, 95% CI 0.76, 1.16) nor PFS (OR 0.96, 95% CI 0.78, 1.18) were associated with HIV

**Table 2. LTBI status, *Mtb*-specific CD4 T-cell activation, and HIV acquisition.**

| CoR Score | Univariate regression estimates | | Multivariate regression estimates | |
|---|---|---|---|---|
| | OR[a] (95% CI) | p-value | aOR[a,b] (95% CI) | p-value |
| LTBI Positive[c] | 0.84 (0.51, 1.39) | 0.51 | 0.85 (0.51, 1.43) | 0.54 |
| PFS[d] | 0.96 (0.78, 1.18) | 0.68 | 0.96 (0.77, 1.20) | 0.73 |
| FS[d] | 0.94 (0.76, 1.16) | 0.57 | 0.95 (0.76, 1.18) | 0.62 |

Unadjusted and adjusted odds ratios for LTBI status, COMPASS polyfunctional scores (PFS) and functional scores (FS) in those who acquired HIV versus controls.

[a]All models used conditional logistic regression accounting for matching variables (treatment arm and country).

[b]Models were fully fit with all variables that were associated with case control status: gender, risk score, and age.

[c]Prespecified as the primary exposure of interested.

[d]ORs are reported for 1 standard deviation change.

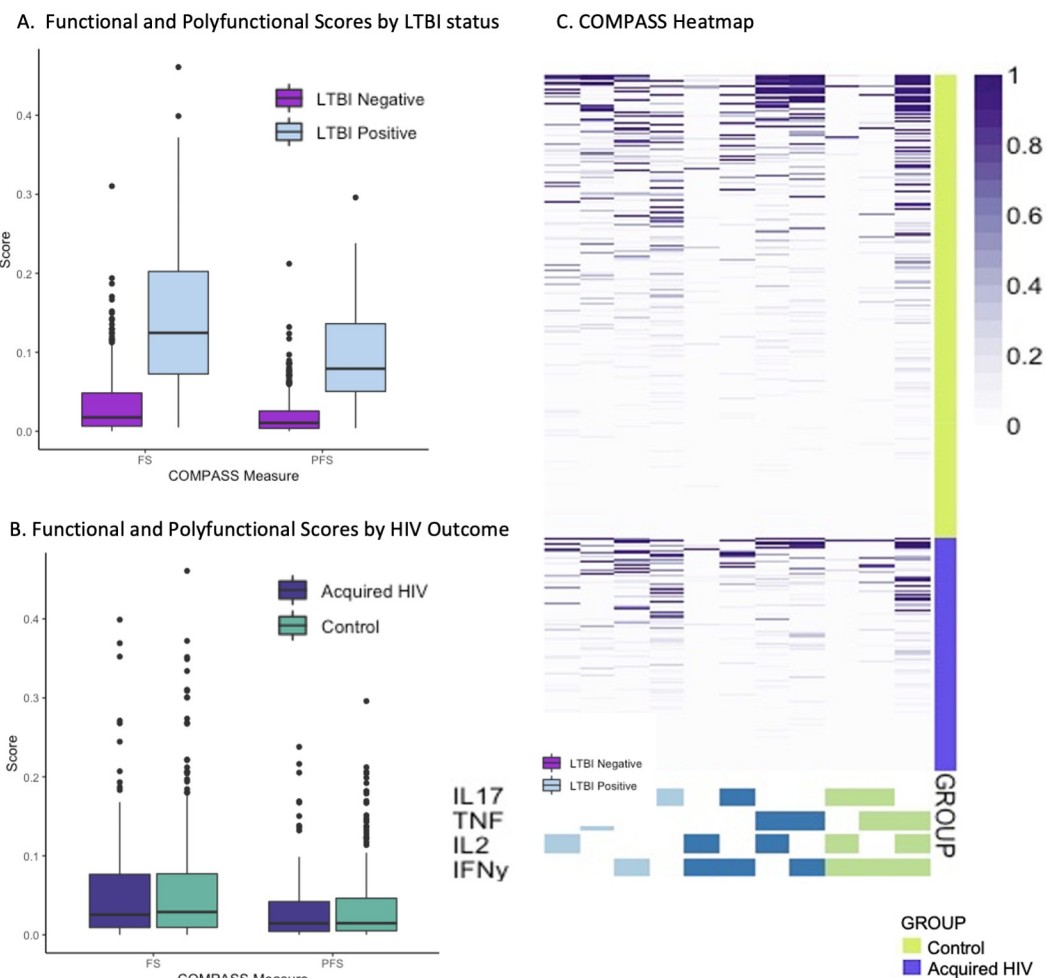

**Fig 1. Combinatorial polyfunctionality analysis of antigen-specific T-cell subsets (COMPASS) describes *Mtb*-antigen-specific T cell activation.** Box and whisker plots depict median and interquartile ranges for FS and PFS scores as well as the range. **A.** COMPASS Polyfunctional Scores (PFS) demonstrate expected higher levels of *Mtb*-specific cellular immune activation in persons with LTBI than those without (p<0.0001). **B.** COMPASS *Mtb*-specific Functional Scores (FS) were no different between persons who acquired HIV and controls. PFS was similarly indistinguishable between cases and controls. **C**. Heatmap of COMPASS posterior probabilities depicts combinations of intracellular cytokine staining composing the combinatorial polyfunctional subsets of *Mtb*-specific CD4+ T cells. Columns show the subsets with detectable antigen-specific responses color-coded by the cytokines they express and ordered by the degree of functionality from one function on the left to 3 of 4 functions on the right; the combinations of cytokines are found in colored boxes at the bottom. Horizontal rows depict one participant, with lime shaded rows representing control samples stacked above violet rows representing persons who acquired HIV. The intensity of purple shading of each box shows the probability that a given participant sample that expresses the subset has a *Mtb*-antigen specific response ranging from white (zero) to purple (one). This heatmap depicts relatively similar distributions of mono- and polyfunctional *Mtb*-specific subsets across persons who did or did not acquire HIV.

acquisition. we found no evidence of an association of FS or PFS with HIV acquisition when evaluating only LTBI positive participants, (**Table C in S1 Appendix**).

RISK6 score was calculated for 439 participants (n = 141 cases, n = 298 controls) in whom RNA extraction was successful. The mean of untransformed RISK6 scores among cases was 0.119 (SD 0.147) compared to 0.107 (SD 0.141) among controls (**Fig 2**). In both unadjusted and adjusted analyses, RISK6 was not associated with HIV acquisition (aOR 1.15 per 1 SD change, 95% CI 0.92, 1.44) (**Table 3**). **Fig 2** shows the distribution of CoR scores in cases who

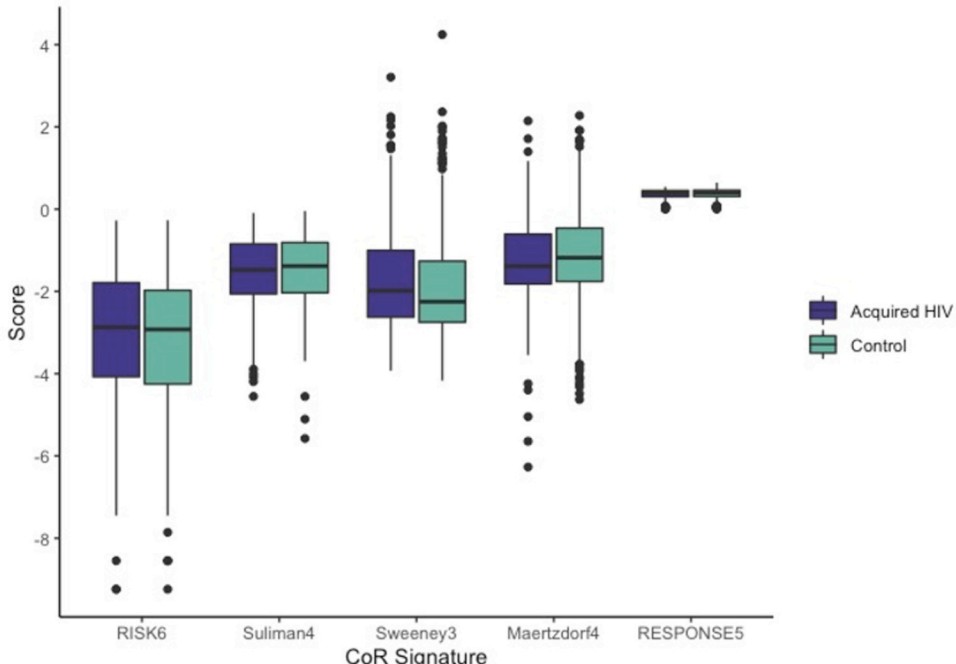

**Fig 2. Distribution of 5 TB Correlates of Risk (CoR) scores among participants who subsequently acquired HIV vs controls.** Box plots demonstrate median and interquartile range of values, with dots completing the range. RISK6 and Suliman4 were $\log_{10}$ transformed as this allowed for the data to be normally distributed and for all scores to be evaluable on a similar scale.

acquired HIV and controls for RISK6, as well as the other four CoR signatures (Suliman4, Sweeney3, Maertzdorf4, and RESPONSE5) examined in exploratory analyses. In multivariate conditional logistic regression, two of the four CoR measures demonstrated associations with HIV acquisition. A one SD increase of the Sweeney3 score was associated with a 38% increased odds of HIV acquisition (aOR 1.38, 95% CI 1.07, 1.78). In contrast, the RESPONSE5 CoR score was associated with a moderate decrease in odds of HIV acquisition (aOR 0.78, 95% CI

**Table 3. Associations between each Correlates of Risk (COR) signature and HIV acquisition, univariate and adjusted analyses.**

| CoR Score[a] | Univariate regression estimates | Multivariate regression estimates |
|---|---|---|
| | OR[c] (95% CI) | aOR[b,c] (95% CI) |
| RISK6[d,e] | 1.06 (0.87, 1.31) | 1.15 (0.92, 1.44) |
| Suliman4[d] | 0.89 (0.72, 1.09) | 0.89 (0.72, 1.11) |
| Sweeney3 | 1.34 (1.05, 1.70) | 1.38 (1.07, 1.78) |
| Maertzdorf4 | 0.90 (0.73, 1.12) | 0.89 (0.71, 1.13) |
| RESPONSE5 | 0.84 (0.67, 1.05) | 0.78 (0.61, 0.98) |

[a]All CoR scores in were modeled using conditional logistic regression accounting for matching variables (treatment arm, country).

[b]OR and adjusted OR (aOR) are reported for 1 standard deviation change.

[c]Models were fully fit with all variables that were associated with case control status: gender, risk score, and age.

[d]Variables log transformed.

[e]Prespecified as the primary transcriptomic signature of interest. The other four signatures were considered exploratory predictors.

0.61, 0.98). There were no important trends in CoR scores between those with lower baseline Ad5 titer, gender, or behavioral risk score. HSV-2 serostatus (in males only) was associated with score of one signature and vaccine vs placebo treatment was associated with a different signature. The mean of three CoR scores differed significantly by country of residence. Bivariate associations between predictors of HIV acquisition and each CoR are shown in **Table D in S1 Appendix**. In adjusted analyses, Ad5 titer was not influential in evaluating associations between any CoR score and HIV acquisition.

Due to the long interval between PBMC collection and HIV diagnosis in some cases, we performed sensitivity analyses including the 282 participants (n = 94 cases, n = 188 controls) with samples within 12 months, and then the 159 participants (n = 53 cases, n = 106 controls) with 6 months or less between sampling and HIV diagnosis. The aOR for RISK6 increased to 1.28 (0.94, 1.75) but neither this or any of the other primary exposures (LTBI, FS, or PFS), became statistically significant for participants sampled within a year or 6 months of diagnosis. As in the full dataset, RESPONSE5 remained associated with decreased risk of HIV and Sweeney3 with the association strengthened by >10% (OR 1.56 (1.10, 2.21)). Due to loss of power, the point estimates for several exploratory analyses changed by >10% for the participants sampled within 6 months, but no findings were significant (**Table E in S1 Appendix**).

In adjusted analyses stratified by LTBI status, a one SD increase in logRISK6 score was associated with a trend toward decreased odds of HIV acquisition among LTBI-positive participants (aOR 0.58, 95% CI 0.34, 0.99) and a trend toward increase in odds of HIV acquisition among LTBI negative participants (aOR 1.24, 95% CI 0.97, 1.59) (**Table F in S1 Appendix**). Stratified by LTBI status, some associations between CoR scores and HIV remained in the LTBI negative participants, but not in the smaller subset of LTBI positive participants. In univariate analyses however, none of the CoR scores were associated with LTBI status (**Table G in S1 Appendix**).

## Discussion

In this study, we accessed rare, cryopreserved specimens from the completed Step HIV vaccine trial to evaluate the influence of sub-clinical *Mtb* infection on HIV acquisition risk, employing a case-control design. Despite the fact that many of the biomarkers that have previously been associated with increased risk of acquiring HIV are also those elevated in subclinical/incipient *Mtb* infection or TB disease, we found little evidence that prior infection with *Mycobacterium tuberculosis* increases risk of acquiring HIV. To our knowledge, there are no cohorts in which LTBI status has been determined using clinical methods (TST or IGRA) among comparable persons with subsequent HIV endpoints. Our use of comprehensive flow cytometry and COMPASS analysis allowed us to retrospectively determine LTBI status and to characterize *Mtb* antigen-specific polyfunctional T-cell responses within a homogenous trial population with subsequent evaluation of HIV status. Neither LTBI status, nor polyfunctional cellular activation scoring was associated with risk of HIV acquisition, either bivariately, or after controlling for the previously described HIV risk factors. Transcriptional TB CoR signatures do not measure *Mtb*-specific responses, but rather the differential expression of genes associated with incipient or active TB disease; we hypothesized that these same signatures, many of which include interferon stimulated genes (ISGs), could also reveal HIV susceptibility states. However, the primary CoR signature of interest, the RISK6 signature, comprising several ISGs, was not associated with HIV acquisition in the overall study. Although aORs differed by LTBI status, as RISK6 was associated with lower HIV risk in those with LTBI and showed a trend towards increased HIV risk in those without LTBI, we could not rule out that this difference was due to chance. It is possible that pathways other than the specific ISG signaling

characterized by RISK6 are associated with HIV risk and are also differentially regulated by co-infections. In addition, evaluation of other CoR profiles suggested associations with HIV acquisition, suggesting that there may be an unmeasured source of immune activation predictive of HIV acquisition. The different gene sets in each signature describe different activation patterns, and possibly various underlying exposures. Because LTBI status was not independently associated with any of the 5 CoRs in this cohort, especially RISK6, Suliman4, or Maetzdorf4, it is likely that there was little incipient TB in the Step study. It is therefore possible that these CoRs were detecting ISGs provoked by stimuli other than *Mtb* infection. The *Mtb*-specific cellular responses that define LTBI are well-known to persist despite LTBI treatment or spontaneous clearance of *Mtb* infection [40, 41], and therefore persons identified as LTBI in this study may not have had differential ISG or other gene expression measured concurrently in that sample or at the time of HIV exposure.

The results of the exploratory analyses provide some evidence that a transcriptomic biomarker could evaluate HIV risk, regardless of *Mtb* infection. The Sweeney3 signature was most strongly associated with HIV risk in unadjusted and adjusted analyses, and in the LTBI-negative, but not the LTBI-positive group. This three-gene set predicts the presence of active TB disease versus LTBI with better accuracy (86% sensitivity, 86% specificity) than most combinations of clinical prediction tools, and with similar sensitivity to GeneXpert-MTB RIF sputum tests [42, 43]. Because the three genes, GBP5, DUSP3, and KLF2, are associated with macrophage regulation and other immune pathways [44–46], this gene set may detect a high-risk systemic immune status, provoked by genital or gastrointestinal infection or altered microbiome composition, which could also be differentially associated with LTBI status. Alternatively, the transcription of these genes could relate to other stable host factors that impact both *Mtb* and HIV susceptibility.

In contrast, the RESPONSE5 profile evaluates differential gene expression that predicts cure after TB treatment among those with active TB, wherein a high score represents higher innate immune activation [28], and in this study, higher scores were associated with lower odds of HIV acquisition. These findings suggest that *Mtb* infection and polyfunctional *Mtb* antigen-specific T-cell activation are not associated directly with HIV risk, but that other unmeasured exposures or intrinsic factors may ultimately activate these same host transcriptional pathways and explain the observed association between some transcriptional signatures and HIV acquisition. For example, some risk scores were modestly associated with receipt of the Ad5 HIV DNA vaccine in the parent trial, with HSV-2 serostatus, or with country of residence.

Our study has several limitations, the foremost of which was lack of availability of stored PBMC close to the time of HIV acquisition in some cases, which we tried to address through sensitivity analyses restricted to cases with proximal samples. Some samples were collected more than one year prior to documented HIV diagnosis. Due to wide intervals between study visits (up to 6 months between visits), many persons likely acquired HIV much closer to the sampling timepoint than indicated by their diagnosis visit, but the timing of HIV acquisition is not clearly pinpointed. That said, CoR signatures are able to predict progression to incident TB up to 24 months prior to clinical disease and show accurate prediction within 1 year [17, 26]. As the parent Step study occurred within some countries with moderate or high TB prevalence, there is a possible risk that interval *Mtb* infection or sub-clinical reactivation occurred between sampling and HIV outcome. Additionally, LTBI status was not determined in the Step study; thus, we were unable to use clinically accepted measures of LTBI status, such as IGRA or TST. Having access to a single blood sample limited our ability to assess interval LTBI status conversion or reversion prior to HIV acquisition, and there is a possibility of other unevaluated infections or other confounders. Nonetheless, the Step study represents a rare

opportunity to address this issue as there are few, if any, cohorts in which *Mtb* infection status, future HIV outcome, and longitudinal cryopreserved specimens are available. Instead, we used a rigorous *Mtb*-antigen-specific flow cytometry assay that provided richer descriptive data, not only about LTBI status, but about the degree and quality of *Mtb* responses. Other strengths of this study include rigorous characterization of study participants as part of the parent Step study, and novel application of TB CoR signatures.

Based on this translational study we would not expect that treating LTBI in persons at risk for HIV would decrease risk of HIV acquisition substantially, especially in the context of increasingly effective and available HIV pre-exposure prophylaxis (PrEP) options. Treating LTBI has other intrinsic benefits, including prevention of TB disease and *Mtb* transmission, especially in those who are living with HIV or otherwise at high risk of developing active TB disease.

To our knowledge there is little published data on differential gene regulation being associated with HIV susceptibility or protection. While we found variable associations with some purpose-designed TB CoR signatures, global transcriptomic analysis could assist in discovery of similar, simpler HIV CoR signatures. Although such HIV CoR signatures would be unlikely to replace patient-reported indications for PrEP and other prevention measures, such signatures could be beneficial in risk-stratifying participants within HIV prevention and vaccine studies. Transcriptomic signatures could also be employed to detect high-risk post-vaccine immune responses in early-phase trial volunteers. Designing purpose-built signatures could help identify vaccine candidates that might induce risky immune states for HIV acquisition, such as occurred in the Step study, before moving the vaccine candidate to larger trials in HIV-exposed populations.

## Conclusions

In this study, we found no evidence that LTBI or polyfunctional Mtb-antigen-specific T cell responses were associated with risk of HIV acquisition in a population at high risk for HIV. However, the exploratory analyses provided a proof of concept that transcriptomic signatures could provide additional information about immunologic profiles of HIV susceptibility beyond known behavioral and viral co-infection risk factors identified in the Step study and other prior studies.

## Supporting information

**S1 Appendix. Supplemental Tables A and B provide materials used in flow cytometry and transcriptional signature analysis, respectively.** Tables C, E, and F provide sensitivity analyses stratified or restricted by LTBI status (C, F) and time restricted analyses (E). Table D shows univariate analyses between predictors of HIV acquisition and CoR scores. Table G shows univerate associate.
(DOCX)

## Acknowledgments

We thank the Step study volunteers and the study staff for their contributions, especially Ashley Clayton, Lisa Bunts, and Todd Haight for facilitating HVTN ancillary studies. We thank Dr. Michael Robertson and Merck Research Laboratories for their contributions to the Step study. We acknowledge Erik Layton and Chetan Seshadri for technical assistance with flow cytometry.

This work has been presented in part at the 2020 CFAR national meeting, virtually in San Diego, USA Nov 4, 2020.

## Author Contributions

**Conceptualization:** Rachel A. Bender Ignacio, Thomas J. Scriba, Mark Hatherill, Gavin Churchyard, Ann Duerr, Javeed A. Shah, Thomas R. Hawn.

**Data curation:** Jessica Long.

**Formal analysis:** Jessica Long, Ethan Valinetz.

**Funding acquisition:** Rachel A. Bender Ignacio, Ann Duerr, Thomas R. Hawn.

**Investigation:** Rachel A. Bender Ignacio, Aparajita Saha, Felicia K. Nguyen, Lara Joudeh, Susan Buchbinder, Ann Duerr, Javeed A. Shah, Thomas R. Hawn.

**Methodology:** Rachel A. Bender Ignacio, Jessica Long, Ethan Valinetz, Simon C. Mendelsohn, Thomas J. Scriba, Mark Hatherill, Holly Janes, Gavin Churchyard, Javeed A. Shah, Thomas R. Hawn.

**Project administration:** Rachel A. Bender Ignacio, Felicia K. Nguyen, Thomas R. Hawn.

**Resources:** Rachel A. Bender Ignacio, Susan Buchbinder, Ann Duerr, Javeed A. Shah, Thomas R. Hawn.

**Supervision:** Rachel A. Bender Ignacio, Thomas J. Scriba, Mark Hatherill, Holly Janes, Gavin Churchyard, Ann Duerr, Javeed A. Shah, Thomas R. Hawn.

**Validation:** Rachel A. Bender Ignacio, Aparajita Saha, Lara Joudeh, Simon C. Mendelsohn, Javeed A. Shah.

**Writing – original draft:** Rachel A. Bender Ignacio, Thomas R. Hawn.

**Writing – review & editing:** Rachel A. Bender Ignacio, Simon C. Mendelsohn, Thomas J. Scriba, Mark Hatherill, Holly Janes, Gavin Churchyard, Susan Buchbinder, Ann Duerr, Javeed A. Shah, Thomas R. Hawn.

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
