## [Decision Letter · Decision Letter 0]

6 Mar 2022

PONE-D-21-33120Mycobacterium tuberculosis infection, immune activation, and risk of HIV acquisitionPLOS ONE

Dear Dr. Bender Ignacio,

Thank you for submitting your manuscript to PLOS ONE. After careful consideration, we feel that it has merit but does not fully meet PLOS ONE’s publication criteria as it currently stands. Therefore, we invite you to submit a revised version of the manuscript that addresses the points raised during the review process.

Please pay attention to the reviewers comments especially in regards to restricting the conclusion mainly to the primary analysis that pre-existing TB exposure does not associate with subsequent HIV acquisition. Please ensure that your decision is justified on PLOS ONE’s publication criteria and not, for example, on novelty or perceived impact.

We look forward to receiving your revised manuscript.

Kind regards,

Manish Sagar, MD

Academic Editor

PLOS ONE

Journal Requirements:

Reviewers' comments:

Reviewer's Responses to Questions

**Comments to the Author**

1. Is the manuscript technically sound, and do the data support the conclusions?

Reviewer #1: Yes

Reviewer #2: Partly

Reviewer #3: Yes

2. Has the statistical analysis been performed appropriately and rigorously? 

Reviewer #1: Yes

Reviewer #2: Yes

Reviewer #3: Yes

3. Have the authors made all data underlying the findings in their manuscript fully available?

Reviewer #1: Yes

Reviewer #2: Yes

Reviewer #3: No

4. Is the manuscript presented in an intelligible fashion and written in standard English?

Reviewer #1: Yes

Reviewer #2: Yes

Reviewer #3: Yes

5. Review Comments to the Author

Reviewer #1: This nested case-control study used prospectively collected data to investigate whether Tuberculosis-associated immunological variables measured in PBMC samples could predict HIV acquisition. The methods are in general appropriate and clearly reported.

The Benjamini-Hochberg methods using false discovery rates (FDR) was developed for the case where very large numbers of significance tests are performed, in the absence of structure or hierarchy in the set of tests. I am not sure whether it is so appropriate in this study: for the main analysis there seem to have been 2 tests (univariate, multivariate) for measure (1) =LTBI, 4 for measure (2) =FS, PFS and 10 tests for measure (3) =CoR) – 16 in total. Sensitivity analyses on subsets defined by sample collection date resulted in a further 2 x 16 tests, while stratification by LBTI status produced 10 further tests. Other significance tests (supplement) were used to investigate correlations between measures, but not association with HIV, so these should presumably not be counts in the BH procedure?

A suggestion would be to restrict the multiple testing adjustment to the main analyses, defining either univariate or multivariate regression as primary (i.e. 8 primary tests), and to describe all other analyses as sensitivity analyses and the corresponding tests as purely descriptive.

In any case, for clarity, the number of tested hypotheses and the preset FDR (=0.05?) should be stated. In order to allow the reader to apply alternative multiple testing approaches, it would be helpful to report the nominal p-values as well as the adjusted ones.

Minor points

1. Please describe more precisely how matched controls were selected for cases with more than two potential controls – using random numbers? How was the balance in pooled gender distribution maintained?

2. P.7 missing word(s) in ‘which predicts cure of at time of TB treatment initiation’?

3. What was the reasoning behind stratification by LTBI status for analyses of the association of PFS, FS, and CoR scores with HIV acquisition (p.7).

4. Were confidence intervals also adjusted using FDR methods as described by Benjamini et al. (reference 40)?

5. On p.10 the mean and standard deviation are given for FS and PFS, although Fig 1 boxplots show very skew distributions. Therefore, median and interquartile range would be more informative.

6. The legend to Fig. 1 seems to state the wrong labels 1C and 1D (heatmap is 1C, there is no 1D).

7. In Fig. 2, the log transformation does not seem to have resulted in a normal (symmetrical) distribution for RESPONSE5 (many extreme negative log values).

Reviewer #2: The authors leverage a unique cohort sample bank to investigate association of tuberculosis (TB) infection and incipient disease with HIV acquisition. Strengths of the manuscript and project are the longitudinal data and access to samples prior to HIV infection. The stronger finding is that TB infection, as defined by the team, was not associated with higher HIV infection risk. The team though makes leaps between signature patterns and interpreting their meaning that seems beyond the associations they can more accurately describe. Other weakness includes lack of TB infection testing in host (which the team acknowledges) and long intervals between pre-HIV blood sample and HIV infection. I believe the manuscript needs to address the following issues.

• In the abstract, describe the cohort at high risk for HIV acquisition (not assume knowledge of what the Step MRKAd5 HIV-1 study included). Last sentence in the conclusion that host gene expression is associated with HIV acquisition feels the wrong focus, instead that pattern is associated with higher likelihood of infection is what I think is captured.

• Throughout the manuscript, concerned saying that a signature predicts HIV acquisition, given there are many mediating steps including need for exposure which will vary by participant. Instead would stick with associated with, with I think main hypothesis which the signature indicates a potential increased vulnerability to successful infection.

• Multiple transcript signatures were tested, with 1 associated with a significant increased odds and another with a lower odds. Would benefit from a clearer indication of what was considered a cutoff with each score that is meaningful and what number of participants were in the “incipient” TB group by each score. Were there enough in that category to have captured their particular risk?

• The discussion appropriately focuses on lack of association with TB infection and HIV infection in this cohort. But the leap that transcriptional signatures might be revealing HIV susceptibility is not well supported, as that is not what these signatures were developed for. Can consider future development but seems inappropriate to interpret from this data.

• Would keep focus on paper as this is exploratory, not making major decisions such as role of TB preventive therapy.

Reviewer #3: Bender Ignacio et al takes advantage of the Step MRKAd5 HIV-1 vaccine study to use samples from a cohort in which Mtb status and approximate time of HIV infection are known to determine if several risk factors associated with latent TB infection or immune activation correlated with HIV infection. In general, no differences were observed and latent TB infection did not appear to be a predictive risk for HIV-1. Even though the data are “negative,” this study, despite the potential caveats pointed out by the authors, is one of the few studies using human samples providing a useful reference for others examining infections in other cohorts. Some minor criticisms include:

1. Page 10, there is a reference to data not shown. These data should be included if not in the text, then in the supplemental data.

2. This is not an easy paper to read; there is an "alphabet soup" of clinical measurements that at times detract from the overall conclusions. Its clinical focus on the outcomes and statistical analysis rather than what the assays are actually measuring limits the appeal to what may be a more general audience. Defining and discussing the immune and cellular signatures in the context of the results, would put into perspective of why these are relevant correlates for TB and HIV and provide a foundation for those with broader interests in immunology, TB, HIV and coinfections. This is also an opportunity to expand the discussion and put into the context with the body of work that has suggested cytokine and functional immune changes.

6. PLOS authors have the option to publish the peer review history of their article (what does this mean?). If published, this will include your full peer review and any attached files.

Reviewer #1: **Yes: **Jeremy Franklin

Reviewer #2: No

Reviewer #3: No

---

## [Author Response · Author response to Decision Letter 0]

5 Apr 2022

Point by point response to reviewers:

Reviewer #1: This nested case-control study used prospectively collected data to investigate whether Tuberculosis-associated immunological variables measured in PBMC samples could predict HIV acquisition. The methods are in general appropriate and clearly reported.

The Benjamini-Hochberg methods using false discovery rates (FDR) was developed for the case where very large numbers of significance tests are performed, in the absence of structure or hierarchy in the set of tests. I am not sure whether it is so appropriate in this study: for the main analysis there seem to have been 2 tests (univariate, multivariate) for measure (1) =LTBI, 4 for measure (2) =FS, PFS and 10 tests for measure (3) =CoR) – 16 in total. Sensitivity analyses on subsets defined by sample collection date resulted in a further 2 x 16 tests, while stratification by LBTI status produced 10 further tests. Other significance tests (supplement) were used to investigate correlations between measures, but not association with HIV, so these should presumably not be counts in the BH procedure?

A suggestion would be to restrict the multiple testing adjustment to the main analyses, defining either univariate or multivariate regression as primary (i.e. 8 primary tests), and to describe all other analyses as sensitivity analyses and the corresponding tests as purely descriptive.

In any case, for clarity, the number of tested hypotheses and the preset FDR (=0.05?) should be stated. In order to allow the reader to apply alternative multiple testing approaches, it would be helpful to report the nominal p-values as well as the adjusted ones.

Thank you for this discussion. We had previously erred conservatively by adjusting all presented analyses, even though the primary hypotheses are few, and the majority of the paper, as pointed out by this reviewer, was done as exploratory testing to evaluate further associations after the primary hypotheses were found null. We have two primary hypotheses in this paper: 1. whether LTBI, measured by the frequency of IFN-ɣ+ CD4+ cells stimulated with ESAT-6/CFP-10, is associated with HIV acquisition, or 2. Whether the RISK6 CoR transcriptional signature is associated with HIV acquisition. We also examine whether several exploratory variables including flow cytometry (COMPASS derived FS and PFS scores) and transcriptional (Suliman4, Maertzdorf4, Sweeny3, RESPONSE5) measurements are associated with HIV acquisition. We revised the manuscript to more clearly distinguish these primary from secondary hypotheses and analyses. For example, we added a horizontal cut to Tables 2 and 3 to distinguish the primary analyses on top from exploratory below the line in both cases.

We had previously only adjusted for the number of tests within a given hypothesis set but had done this separately for each set of data presented (results from flow cytometry as a separate set from transcriptomic signatures, which resulted in correction for only a few comparisons, except in the supplementary tables). We did not count univariate and multivariate analyses as separate tests since they provide different types of information about the same hypothesis test. For a similar reason, we feel that providing both the nominal and BH adjusted p values puts disproportionate emphasis on the significance test, rather than the effect size and direction, and the inference that can be gained by comparing adjusted and unadjusted tests of the primary hypotheses. 

Therefore, given that there are only two primary hypotheses tested (LTBI or RISK6 association with HIV risk), we are no longer presenting unadjusted p values for the flow analyses in Table 2, which includes only 3 tests (1 primary and 2 secondary), with no change in inference with either strategy (none approached significance with nominal p-values). For the CoR scores, we had prespecified RISK6 as the primary exposure, and therefore now present RISK6 as the single primary signature of interest and more clearly specify that the other 4 scores were done as exploratory analyses. Again, the nominal p value for RISK6 did not approach 0.05, so we are embracing the null hypothesis with either method. 

At the suggestion of the reviewer, we have reverted to nominal p values for the exploratory analyses in the supplement or else do not display p values, as the overarching goal of these additional analyses was to identify plausible explanations for which known clinical or demographic risk factors for HIV, could be associated with immune responses that could render a person more susceptible to HIV if Mtb itself was not a risk factor. 

Minor points

1. Please describe more precisely how matched controls were selected for cases with more than two potential controls – using random numbers? How was the balance in pooled gender distribution maintained? 

The HVTN laboratory team offered controls based on availability of PBMCs with the matched visit to the cases selected, of those who still had valid consent for future use. The list was generated by HVTN statisticians based on matching requirements, and since they were not able to be directly matched on gender, they selected roughly the same number of female and male controls as in cases. The HVTN laboratory did this before providing the entire transfer of samples through their ancillary studies mechanism, which arranges receipt of samples and a matching dataset for external or collaborating investigators. 

An unblinded team member not involved in the laboratory work constructed a batching scheme that maintained a rough 2:1 proportion of cases and controls for each planned plate but then stripped these details from the sample manifest. These steps allowed for balanced selection of controls to cases throughout the lab work, and for our laboratory team to remain blinded to case/control status and the other clinical/demographic details as they ran the samples. 

We have added more detail in the methods to this point. 

2. P.7 missing word(s) in ‘which predicts cure of at time of TB treatment initiation’? 

Addressed, thank you.

3. What was the reasoning behind stratification by LTBI status for analyses of the association of PFS, FS, and CoR scores with HIV acquisition (p.7).

The stratified analysis was initially planned as a mediation analysis to evaluate what proportion of effect of LTBI on HIV incidence was attributable to the pathway of transcriptional activation through RISK6 (or another score). However, LTBI was not associated with incident HIV, but some of the CoRs were. So, we then performed this analysis to explore additional hypotheses as to why some CoRs could be associated with HIV in the absence of LTBI signal: 

1) Whether CoRs were associated with HIV, but only among people with LTBI. If the effect size was strengthened or only present in the LTBI subset, compared to the overall group, that could have indicated that LTBI alone was not sufficient to increase risk of HIV, but that Mtb-associated immune activation was. This was one of our primary hypotheses- that being infected with Mtb wasn’t sufficient, and incipient TB was needed to see the association with HIV acquisition. We did not find this.

2) If the CoR-> HIV pathway was only seen in people without LTBI, then maybe there was a different unmeasured exposure that was associated with HIV acquisition among those without LTBI. This would require something else to be different among people without LTBI related to social status, exposures, sexually transmitted infections, which is possible. We therefore looked at whether HSV status, Ad5 titer, or treatment arm, for example were associated with CoRs in univariate analyses (Supplementary Table 4) in case we could identify a risk driver that was not LTBI. We found some hints of this, and therefore one conclusion is that while LTBI is not a unique risk, one might consider making a purpose-built CoR for HIV risk, and then further exploring associations with other non-Mtb exposures that drive them. 

We have tried to add more of this into the Discussion, as we de-emphasized the interpretation of those CoR findings, as suggested by Reviewer 2.

4. Were confidence intervals also adjusted using FDR methods as described by Benjamini et al. (reference 40)?

No. We present the original confidence intervals. See longer discussion above, but we now present unadjusted p values also for the 3 flow results. 

5. On p.10 the mean and standard deviation are given for FS and PFS, although Fig 1 boxplots show very skew distributions. Therefore, median and interquartile range would be more informative.

Thank you for this suggestion. We have made this change. 

6. The legend to Fig. 1 seems to state the wrong labels 1C and 1D (heatmap is 1C, there is no 1D).

This has been corrected.

7. In Fig. 2, the log transformation does not seem to have resulted in a normal (symmetrical) distribution for RESPONSE5 (many extreme negative log values).

We had previously chosen the log transformation so as to bring the range of scores closer to other CoRs and to create a better spread in the data as the range is very narrow. However, we also appreciate this point, and now present RESPONSE5 as neat rather than log transformed throughout. 

Reviewer #2: The authors leverage a unique cohort sample bank to investigate association of tuberculosis (TB) infection and incipient disease with HIV acquisition. Strengths of the manuscript and project are the longitudinal data and access to samples prior to HIV infection. The stronger finding is that TB infection, as defined by the team, was not associated with higher HIV infection risk. The team though makes leaps between signature patterns and interpreting their meaning that seems beyond the associations they can more accurately describe. Other weakness includes lack of TB infection testing in host (which the team acknowledges) and long intervals between pre-HIV blood sample and HIV infection. I believe the manuscript needs to address the following issues.

We appreciate this assessment. We also note that while the parent trial did not include either PPD or clinically used Interferon Gamma Release Assays (IGRAs), the flow cytometry assessment in this study using ESAT-6 and CFP-10 uses these same antigens found in authorized IGRA tests and has been shown to perform similarly to IGRAs in the hands of the authors and as published by several others (Refs 31-36). There are no tests for current TB infection other than sputum analysis for M. tuberculosis DNA or growth, which only identifies active TB; the IGRA tests or this analogous flow cytometry version are the most specific available tests for Mtb exposure.

• In the abstract, describe the cohort at high risk for HIV acquisition (not assume knowledge of what the Step MRKAd5 HIV-1 study included). 

We clarified that the Step study was a previously completed multinational HIV vaccine study. 

Last sentence in the conclusion that host gene expression is associated with HIV acquisition feels the wrong focus, instead that pattern is associated with higher likelihood of infection is what I think is captured.

Thank you, we’ve tried to make the wording clearer throughout the discussion, including moving some of the sentences from the conclusion to the part of the discussion that expounds on exploratory analyses and next steps. We have followed this reviewers’ suggestion and now end with a more succinct summary of the study, as follows:

In this study, we found no evidence that LTBI or polyfunctional Mtb-antigen-specific T cell responses were associated with risk of HIV acquisition in a population at high risk for HIV. However, the exploratory analyses provided a proof of concept that transcriptomic signatures could provide additional information about immunologic profiles of HIV susceptibility beyond known behavioral and viral co-infection risk factors identified in the Step study and other prior studies. 

• Throughout the manuscript, concerned saying that a signature predicts HIV acquisition, given there are many mediating steps including need for exposure which will vary by participant. Instead would stick with associated with, with I think main hypothesis which the signature indicates a potential increased vulnerability to successful infection.

We have made edits throughout to use language that does not imply causality. In some places, we left this nomenclature where the word “predict” was appropriate, as in discussing a future event in longitudinal analysis. 

• Multiple transcript signatures were tested, with 1 associated with a significant increased odds and another with a lower odds. Would benefit from a clearer indication of what was considered a cutoff with each score that is meaningful and what number of participants were in the “incipient” TB group by each score. Were there enough in that category to have captured their particular risk?

We used these TB CoRs as continuous variables in all cases, whether log-transformed or neat. Because these CoRs are not tuned as HIV risk scores, we used them without assumptions of cutoffs, which also gave us the full power of continuous rather than binary predictors. For example, even if there is a published cutoff for the TB status of interest (which is incipient TB in only 3 scores, LTBI vs active TB in 1 score (Sweeney3), and prediction of TB cure at treatment start in the last score (RESPONSE5)), we did not make assumptions as to what level of immune activation would be associated with HIV acquisition. 

We present OR throughout for 1 SD change of each score to use these scores in an unbiased way. 

The fact that the 5 scores predict different TB statuses is likely why some scores are positively associated and other negatively associated with HIV. We have tried to clarify this more with what the scores represent in the discussion. For example, RESPONSE5 is likely negatively associated with HIV risk because this score predicts successful TB cure at the start of treatment, rather than associated with incipient TB.

We also have tried to more clearly identify that RISK6 was the primary transcriptomic signature of interest, with the rest as exploratory exposure variables. 

• The discussion appropriately focuses on lack of association with TB infection and HIV infection in this cohort. But the leap that transcriptional signatures might be revealing HIV susceptibility is not well supported, as that is not what these signatures were developed for. Can consider future development but seems inappropriate to interpret from this data. 

We have tried to now state this more carefully. What we concluded was not that we should use any of these signatures to predict HIV acquisition, but rather that the exploratory work we did gives credence to the idea that an HIV purpose-built CoR could be designed. We would encourage colleagues to use unbiased approaches (eg high-throughput sequencing) to identify transcriptomic signatures that could be similarly used for HIV.

To address this point, we added the following text to the discussion to call for purpose-built signatures for HIV risk developed via unbiased approaches: “While we found variable associations with some purpose-designed TB CoR signatures, global transcriptomic analysis could assist in discovery of similar, simpler HIV CoR signatures.”

And have revised the discussion as follows: “However, the exploratory analyses provided a proof of concept that transcriptomic signatures could provide additional information about immunologic profiles of HIV susceptibility”

• Would keep focus on paper as this is exploratory, not making major decisions such as role of TB preventive therapy. 

We agree and are not advocating for TB preventative therapy in our conclusion. We discuss the idea that TB prevention could have been considered to avert HIV infections in populations with HIV risk in TB-endemic areas worldwide, as this was an actionable public health intervention that motivated us to pursue this project. Because we did not find an association with LTBI or flow or transcriptomic markers of TB-associated immune activation, we cannot justify pursuing a clinical trial with this aim, which would have been the next step had our analyses shown a strong association. Because our primary hypotheses were negative, we did perform exploratory analyses evaluating whether LTBI or CoRs were associated with other known HIV risk factors in the parent trial. 

Reviewer #3: Bender Ignacio et al takes advantage of the Step MRKAd5 HIV-1 vaccine study to use samples from a cohort in which Mtb status and approximate time of HIV infection are known to determine if several risk factors associated with latent TB infection or immune activation correlated with HIV infection. In general, no differences were observed and latent TB infection did not appear to be a predictive risk for HIV-1. Even though the data are “negative,” this study, despite the potential caveats pointed out by the authors, is one of the few studies using human samples providing a useful reference for others examining infections in other cohorts. Some minor criticisms include:

1. Page 10, there is a reference to data not shown. These data should be included if not in the text, then in the supplemental data. 

Thank you for this suggestion. We added this data as a new Supplementary Table 3 and renumbered the subsequent Supplemental tables. 

2. This is not an easy paper to read; there is an "alphabet soup" of clinical measurements that at times detract from the overall conclusions. Its clinical focus on the outcomes and statistical analysis rather than what the assays are actually measuring limits the appeal to what may be a more general audience. Defining and discussing the immune and cellular signatures in the context of the results, would put into perspective of why these are relevant correlates for TB and HIV and provide a foundation for those with broader interests in immunology, TB, HIV and coinfections. This is also an opportunity to expand the discussion and put into the context with the body of work that has suggested cytokine and functional immune changes.

We have tried to improve the language used in the discussion to make it more easily interpretable. 

For example, we added more detail to how the 5 CoRs are used for TB prediction, so that they are not simply acronyms. We also completely rewrote a large section of the discussion and conclusion (see response to Reviewer #2) in an attempt to make it easier to follow.

Thank you for the opportunity to respond to these comments.

---

## [Editor Report · Decision Letter 1]

14 Apr 2022

Mycobacterium tuberculosis infection, immune activation, and risk of HIV acquisition

PONE-D-21-33120R1

Dear Dr. Bender Ignacio,

We’re pleased to inform you that your manuscript has been judged scientifically suitable for publication and will be formally accepted for publication once it meets all outstanding technical requirements.

Kind regards,

Manish Sagar, MD

Academic Editor

PLOS ONE

---

## [Editor Report · Acceptance letter]

21 Apr 2022

PONE-D-21-33120R1 

*Mycobacterium tuberculosis* infection, immune activation, and risk of HIV acquisition 

Dear Dr. Bender Ignacio:

I'm pleased to inform you that your manuscript has been deemed suitable for publication in PLOS ONE. Congratulations! Your manuscript is now with our production department. 

Kind regards, 

on behalf of

Dr. Manish Sagar 

Academic Editor

PLOS ONE